# Nationwide germline whole genome sequencing of 198 consecutive pediatric cancer patients reveals a high incidence of cancer prone syndromes

**Anna Byrjalsen**[1,2], **Thomas V. O. Hansen**[1,2], **Ulrik K. Stoltze**[1], **Mana M. Mehrjouy**[1,2], **Nanna Moeller Barnkob**[3], **Lisa L. Hjalgrim**[2], **René Mathiasen**[2], **Charlotte K. Lautrup**[4], **Pernille A. Gregersen**[5], **Henrik Hasle**[6], **Peder S. Wehner**[7], **Ruta Tuckuviene**[8], **Peter Wad Sackett**[3], **Adrian O. Laspiur**[3], **Maria Rossing**[9], **Rasmus L. Marvig**[9], **Niels Tommerup**[10], **Tina Elisabeth Olsen**[11], **David Scheie**[11], **Ramneek Gupta**[3], **Anne–Marie Gerdes**[1‡], **Kjeld Schmiegelow**[2,12‡], **Karin Wadt**[1‡]*

1 Department of Clinical Genetics, Copenhagen University Hospital Rigshospitalet, Copenhagen, Denmark, 2 Department of Paediatrics and Adolescent Medicine, Copenhagen University Hospital Rigshospitalet, Copenhagen, Denmark, 3 Department of Health Technology, Technical University of Denmark, Copenhagen, Denmark, 4 Department of Clinical Genetics, Aalborg University Hospital, Aalborg, Denmark, 5 Department of Clinical Genetics, Aarhus University Hospital, Aarhus, Denmark, 6 Department of Paediatrics and Adolescent Medicine, Aarhus University Hospital, Aarhus, Denmark, 7 Department of Paediatric Hematology and Oncology, H. C. Andersen Children's Hospital, Odense University Hospital, Odense, Denmark, 8 Department of Paediatrics and Adolescent Medicine, Aalborg University Hospital, Aalborg, Denmark, 9 Center for Genomic Medicine, Copenhagen University Hospital Rigshospitalet, Copenhagen, Denmark, 10 Department of Cellular and Molecular Medicine, University of Copenhagen, Copenhagen, Denmark, 11 Department of Pathology, Copenhagen University Hospital Rigshospitalet, Copenhagen, Denmark, 12 Institute of Clinical Medicine, Faculty of Health and Medical Sciences, University of Copenhagen, Copenhagen, Denmark

‡ These authors are share senior authors on this work.
* karin.wadt@regionh.dk

**Data Availability Statement:** All relevant data are within the manuscript and its Supporting Information files.

## Abstract

PURPOSE: Historically, cancer predisposition syndromes (CPSs) were rarely established for children with cancer. This nationwide, population-based study investigated how frequently children with cancer had or were likely to have a CPS. METHODS: Children (0–17 years) in Denmark with newly diagnosed cancer were invited to participate in whole-genome sequencing of germline DNA. Suspicion of CPS was assessed according to Jongmans'/McGill Interactive Pediatric OncoGenetic Guidelines (MIPOGG) criteria and familial cancer diagnoses were verified using population-based registries. RESULTS: 198 of 235 (84.3%) eligible patients participated, of whom 94/198 (47.5%) carried pathogenic variants (PVs) in a CPS gene or had clinical features indicating CPS. Twenty-nine of 198 (14.6%) patients harbored a CPS, of whom 21/198 (10.6%) harbored a childhood-onset and 9/198 (4.5%) an adult-onset CPS. In addition, 23/198 (11.6%) patients carried a PV associated with biallelic CPS. Seven of the 54 (12.9%) patients carried two or more variants in different CPS genes. Seventy of 198 (35.4%) patients fulfilled the Jongmans' and/or MIPOGG criteria indicating an underlying CPS, including two of the 9 (22.2%) patients with an adult-onset CPS versus 18 of the 21 (85.7%) patients with a childhood-onset CPS (p = 0.0022), eight of the

**Funding:** This study was financially supported by the Danish Childhood Cancer Foundation (KS, https://boernecancerfonden.dk/), Rigshospitalet (AB, KW, https://www.rigshospitalet.dk/), The Danish Cancer Society (KS), the Foundation for Health Research of the Capital Region of Denmark (AMG, https://www.regionh.dk/til-fagfolk/Forskning-og-innovation/finansiering-og-fonde/s%C3%B8g-regionale-midler/Sider/Region-Hovedstadens-forskningsmidler.aspx), The European Union's Interregional Öresund–Kattegat–Skagerrak grant (KS, https://interreg-oks.eu/), Aase and Ejnar Danielsen's Foundation (AMG, https://danielsensfond.dk/), and Engineer Otto Christensen's Foundation (AB, no URL), the Danish National Research Foundation (RLM, https://dg.dk/, grant number 126). No funding sources played a role in study design, data collection, analysis, decision to publish, or preparation of the manuscript.

**Competing interests:** The authors have declared that no competing interests exist.

additional 23 (34.8%) patients with a heterozygous PV associated with biallelic CPS, and 42 patients without PVs. Children with a central nervous system (CNS) tumor had family members with CNS tumors more frequently than patients with other cancers (11/44, $p = 0.04$), but 42 of 44 (95.5%) cases did not have a PV in a CPS gene. CONCLUSION: These results demonstrate the value of systematically screening pediatric cancer patients for CPSs and indicate that a higher proportion of childhood cancers may be linked to predisposing germline variants than previously supposed.

## Author summary

Traditionally pediatric cancer have been thought to be–mostly–caused by pure bad luck. In recent years, however, this notion has been challenged by novel findings as both maternal environmental exposure and genetic causes have been proven to increase the risk of certain pediatric cancers. With this study we have investigated a national cohort of pediatric cancer patients in Denmark. We have mapped family pedigree, made physical examination of the patients, and sequenced their genome, to get a 360-degree understanding of these patients. This revealed that a tenth of all patients carried a genetic variant causative of their cancer development. In addition, almost half of all patients were suspected of carrying a causative genetic variant based on tools that evaluate type of cancer, physical characteristics and family history. It also showed that tools to predict which patients carried a genetic variant did not identify all patients who in fact carried a genetic variant. Overall, roughly half of all patients were suspected of carrying an underlying genetic cause of their cancer, and a tenth had a verified underlying genetic variant predisposing to cancer pediatric cancer. This could suggest that the amount of pediatric cancer cases attributed to genetic factors may be even higher.

## Introduction

In Europe, 15,000 children (1 in 300) are diagnosed with cancer each year.[1] Cancer can be attributed to genetic predisposition, exposure to carcinogens, and/or random mutations during cell division. Children are exposed to fewer carcinogens than adults.[2,3] Therefore, genetic predisposition and randomly acquired mutations are the major causes of most childhood cancers.

Cancer predisposition syndromes (CPSs) were previously considered rare among pediatric cancer patients, but increasing use of whole-exome sequencing (WES) and whole-genome sequencing (WGS) have identified up to 10% CPS among children, including several cases of CPS for adult-onset cancers not previously associated with childhood CPS. However, most studies investigated selected or single institution cohorts and included patients with specific diagnoses that were frequently associated with CPS.[4–6] Although some studies have included a broader range of pediatric cancer patients, [7–11] there have currently been no nationwide population-based studies. Moreover, most studies have focused on single nucleotide variants (SNVs) and few have included the effects of copy number variants (CNVs).[12]

Many clinical criteria have been developed to identify patients with CPS,[13–18] but these have not been validated in a national cohort.

Here we present the genetic SNV and CNV findings from the first 198 consecutive pediatric cancer patients included in the Danish, prospective, nationwide study Sequencing Tumor And Germline DNA—Implications for National Guidelines (STAGING).

## Methods

### Ethics statement

Ethical approval was obtained through the regional scientific ethical committee (the Ethical Scientific Committees for the Capital Region, H-15016782) and the Danish Data Protection Agency (RH-2016-219, I-Suite no: 04804). All parents/guardians and patients 15 years or older gave formal written consent to participation in this study.

### Inclusion criteria and national setup

CPSs were defined as likely pathogenic or pathogenic variants (PVs) in a gene, predisposing the carrier to childhood- or adult-onset cancer. Between 1 July and 31 December 2016 we included 25 patients in the STAGING pilot study at Rigshospitalet (Copenhagen University Hospital, Denmark). On 1 January 2017, the study was expanded to all four pediatric oncology departments in Denmark. All patients were enrolled before June 2018.

Patients were eligible for inclusion if aged 0–17 years at diagnosis of a primary cancer including benign brain tumors, Langerhans cell histiocytosis (LCH), or myelodysplastic syndrome and parents spoke and read Danish.

Families were provided with written and oral information about the study by a research nurse or oncologist. A PhD student from STAGING (AB) or clinical geneticist provided genetic counselling to families interested in participating. Counselling sessions included pedigree construction (three generations), recording the child's clinical phenotypic features according to McGill Interactive Pediatric OncoGenetic Guidelines (MIPOGG)[18] and Jongmans' criteria[13] (Table 1), and explaining the potential consequences of genetic findings. These consequences included secondary findings, variants of unknown significance (VUS), implications of pathogenic findings associated with CPSs, and subsequent preventive and surveillance measures. Families choosing to enroll in the study were informed that PVs in 'actionable' genes listed by the American College of Medical Genetics and Genomics (ACMG)[19] would be disclosed to them. Families could select information regarding:

1: PVs in ACMG 'actionable' genes.

2: "1" and PVs in 314 known and putative cancer genes. Heterozygous variants in genes with solely recessive inheritance patterns were reported only if further familial genetic testing was warranted, in accordance with clinical guidelines.

3: In addition to "1" and "2", PVs in genes unrelated to CPSs (Table 2). Variants were only reported if clinical consequences were anticipated. Findings in these genes are not presented here.

Pedigrees covering 1st–3rd-generation family members were constructed for all patients. 1st-degree family members were parents and siblings, 2nd-degree family members were uncles/aunts and grandparents, and 3rd-degree family members were cousins, grandparents' siblings and great-grandparents. Cancer diagnoses were verified using unique civil registration numbers, which link family members to medical records, including pathological descriptions of cancer, in the Danish Pathology Data Bank. Living family members gave consent, whereas medical records of deceased family members could be retrieved without consent.

### DNA sampling and sequencing

Genomic DNA was isolated from peripheral blood samples. For patients with hematologic malignancies, blood samples were drawn after remission, otherwise skin biopsies were obtained. Parental blood samples were collected to establish whether variants were paternally or maternally derived or occurred *de novo*.

**Table 1. Tools to identify patients at risk of a cancer predisposition syndrome. Excerpt from the updated Jongmans' criteria[13] and McGill Interactive Pediatric OncoGenetic Guidelines (MIPOGG)[17].**

| | |
|---|---|
| **JONGMANS' CRITERIA** | |
| Criteria 1:<br>Family history (3 generations) | Cancer history in the family:<br>- 2 or more malignancies in family members <18 years of age<br>- Any 1st-degree relative with cancer <45 years of age<br>- 2 or more 1st- or 2nd-degree relatives in same parental lineage with cancer <45 years of age<br>Parents are consanguineous |
| Criteria 2:<br>Neoplasm indicating underlying CPS | E.g.,<br>Hypodiploid ALL, botryoid rhabdomyosarcoma of the urogenital tract, gastrointestinal stromal tumors, retinoblastoma, schwannoma, subependymal giant cell astrocytoma |
| Criteria 3:<br>Tumor analysis suggesting germline predisposition | E.g.,<br>Microsatellite instability in constitutional mismatch repair deficiency, loss of heterozygosity, other mutational signatures |
| Criteria 4:<br>Patient with 2 or more malignancies | Secondary, bilateral, multifocal, or metachronous cancers |
| Criteria 5:<br>Congenital or other phenotypic anomalies | Congenital anomalies (oral clefting, skeletal anomalies, facial dysmorphism)<br>Developmental delay<br>Growth anomalies<br>Skin aberrations (café-au-lait spots, hypopigmentation, sun sensitivity)<br>Immune deficiency |
| Criteria 6:<br>Excessive toxicity related to cancer treatment | This criterion is not well defined and, based on an individual assessment by the pediatric oncologist/researcher, we have chosen not to include this criterion in this paper |
| **MIPOGG CRITERIA** | |
| *Universal criteria* | |
| Anamnestic criteria | - >1 primary tumor<br>- Bilateral/multifocal primary tumors<br>- Dysmorphic features/congenital abnormalities that the clinician deems to be related to cancer predisposition |
| Family anamnestic criteria | - Known cancer predisposition syndrome in the family<br>- Close relative* with cancer <18 years OR a parent/sibling/half-sibling with cancer at <50 years<br>- Close relative* with the same cancer type or same organ affected by cancer at any age<br>- Close relative* with multiple primary tumors |
| *Tumors for direct referral* | |
| Tumors of the central nervous system and ocular tumors | Atypical teratoid rhabdoid tumor, choroid plexus carcinoma, dysplastic cerebellar gangliocytoma, endolymphatic sac tumor, hemangioblastoma, optic pathway glioma, pineoblastoma, pituitary adenoma, retinoblastoma, subependymal giant cell astrocytoma, vestibular schwannoma |
| Renal and neuroblastic tumors | Cystic nephroma, renal angiomyolipoma, renal cell carcinoma, renal rhabdoid tumor |
| Bone and soft-tissue tumors | Desmoid tumor, extrarenal rhabdoid tumor, Gardner fibroma, malignant periphery nerve sheath tumor, nasal chondromesenchymal hamartoma |
| Other tumors | Adrenocortical carcinoma, cardiac rhabdomyoma, colorectal carcinoma, gastrointestinal stromal tumor, hepatoblastoma, medullary thyroid cancer, ovarian Sertoli–Leydig cell tumor, parathyroid tumor, pheochromocytoma, paraganglioma, pleuropulmonary blastoma, trichilemmoma, small cell carcinoma of the ovary of hypercalcemic type, carcinoma of the breast, lung, cervix, uterus, or bladder |

WGS was performed by the Norwegian Sequencing Center (Oslo, Norway) for the pilot study and by the Beijing Genomics Institute (Hong Kong, China) for the national study using the HiSeqX platform (Illumina, San Diego, CA, USA) with paired-end sequencing of

**Table 2. Families could choose to receive one of the following levels of feedback from germline WGS of the affected child.**

| | |
|---|---|
| Level 1 | Information regarding pathogenic or likely pathogenic variants in genes identified by the American College of Medical Genetics[19]. These genes are 'actionable' i.e., there are potential preventive, treatment, or surveillance modalities available. Half of these genes are related to CPS (primary findings); the others are related to cardiac disease, metabolic disorders, or familial hypercholesterolemia (secondary findings). |
| Level 2 | In addition to the genes listed at level 1, information regarding pathogenic or likely pathogenic variants in other known or putative CPS genes (from the list of 314 CPS genes found in S 1A Data). These were considered primary findings. However, if there was no known correlation between the clinical phenotype and the gene in question, the variant was considered a secondary finding. |
| Level 3 | In addition to the genes listed at levels 1 and 2, families would also receive information regarding pathogenic or likely pathogenic variants in other genes not related to CPS (not presented in this paper). These were considered secondary findings. |

150-bp reads and 30× average coverage. Reads were mapped to the hg19 reference genome sequence (GRCh37.p13; RefSeq assembly accession GCF_000001405.25) using BWA version 0.7.12,[20] and biobambam2 version 2.0.27[21] was used to sort and mark duplicate reads. Germline SNVs and indel variants were called with HaplotypeCaller using GATK version 3.8[22] or the DNAseq pipeline (Sentieon, San Jose, CA, USA). VarSeq software (version 2.2.0, Golden Helix, Bozeman, MT, USA) was used to annotate variants. Moreover, filtration was based on read depth $\geq 8$, genotype quality $\geq 20$, and variant allele frequency (VAF) $\geq 0.2$, and sequence ontology was used to exclude intronic and intergenic variants and variants located in the 3' and 5' untranslated region (UTR) prior to evaluation. Integrative Genomics Viewer (IGV, version 2.8.2, Boston, MA, USA) was used to visualize read alignments. Manta and CNVkit were applied for calling larger structural rearrangements. [23,24] All variants were reported according to HGVS nomenclature guidelines.[25] WGS data were filtered for PVs in the 59 ACMG 'actionable' genes and 314 cancer genes (S1 Data). The cancer gene panel was selected from Zhang *et al.*,[7] Rahman,[26] and novel genes recently linked to childhood or adult CPSs. All variants with a minor allele frequency of <1% in any large population (gnomAD) were tabulated. For CPS genes with higher variant frequencies in the general population (e.g., *ATM*, *CHEK2*), a separate filter was used. We did not apply a specific variant filter to identify mosaicism. Variants were assessed by a team of clinical geneticists and molecular biologists based on variant type (e.g., frameshift, nonsense, missense), computational predictions of effect on protein and RNA function (e.g., Combined Annotation Dependent Depletion [CADD], PHRED quality score, ADA splice prediction score),[27] and database searches for published literature on each variant. Moreover, we used Alamut Visual 2.10 to evaluate variants effect on splicing (https://www. interactive-biosoftware.com/alamut-visual/). The effects of variants were considered significant if the scores of at least three programs were reduced $\geq 10\%$ or a strong cryptic acceptor or donor site was generated. Variants were classified as pathogenic (class 5), likely pathogenic (class 4), VUS (class 3), likely benign (class 2), and benign (class 1).[28] Class 4 and 5 variants were designated 'PVs'. Class 3 variants, especially those that potentially matched the child's diagnosis, were further investigated by segregation analysis and splice predictions, and tumor RNA sequencing was used to assess loss-of-heterozygosity (LOH) if tissue was available (Fig 1). In addition, we used the machine-learning tool ORVAL to predict whether combinations of genetic variants were likely to be pathogenic.[29] Variants were discussed at regular multidisciplinary meetings by pediatric oncologists, clinical geneticists, and bioinformaticians. PVs were verified by Sanger or next-generation sequencing before parents were informed.

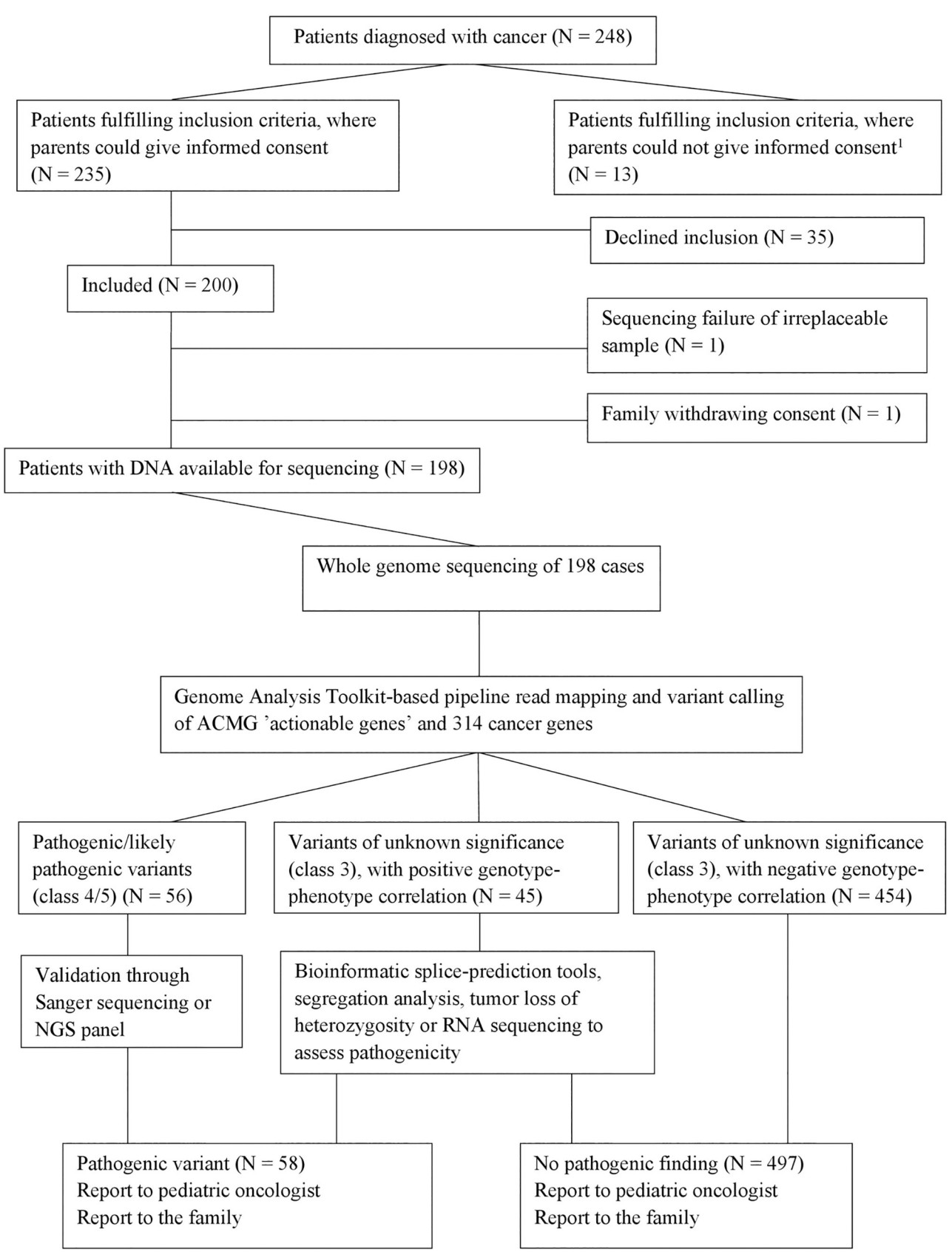

**Fig 1. Inclusion and sequencing strategy. Variants presented are in genes associated with CPS.** All variants of unknown significance have a CADD-PHRED score >20 and an allele frequency <1%. [1]Patients whose parents were not able to give informed consent due to language barriers or social issues (mainly parental psychiatric/severe somatic disease).

## Statistical analysis

Patient/parental characteristics were compared using Pearson chi-square test. Two-sided *P*-values below 0.05 were considered statistically significant. Statistical calculations were carried out using the R software version 3.6.0.

## Results

### Patient characteristics

Of 248 consecutive pediatric cancer patients that fulfilled the inclusion criteria, 198 families consented to participation (Fig 1, Table 3). The included patients did not differ significantly (*p*>0.05) from the 35 patients that declined to participate with regard to sex, age, or diagnoses. Of 198 families, 155/198 (78.3%) opted for feedback on all findings, 24/198 (12.1%) opted for feedback on PVs in 'actionable' genes and 314 cancer genes, and 19/198 (9.6%) opted for feedback on PVs in ACMG 'actionable' genes only. There were no significant differences (*p*>0.05) in parental age, educational level, or income among families opting for the three levels of feedback.

### Patients with known and suspected CPSs

Overall, 94/198 (47.5%) patients carried a PV in a CPS gene or were suspected of having an underlying CPS based on Jongmans'/MIPOGG criteria or family cancer history (Fig 2).

Twenty-nine of 198 (14.6%) patients carried PVs in at least one CPS gene, including four patients with trisomy 21. Of these, 21/198 (10.6%) had PVs in genes predisposing toward childhood-onset CPS, including *CDC73*, *DDX41* (biallelic, putative childhood-onset cancer gene), *LTZR1*, *NF1*, *RB1*, *SDHC*, *SMARCA4*, *TP53*, and *TSC2*, uniparental disomy (UPD) of chromosome 11p (clinical analysis), and trisomy 21.[30]

In addition, 9/198 (4.5%) patients had PVs in genes predisposing toward adult-onset CPS, one of whom also carried a deletion in a childhood-onset CPS gene. Adult-onset CPS variants were identified in *APC*, *ATM*, *AXIN2*, *BARD1*, *BRCA2*, *CHEK2*, *MUTYH*, and *PALB2*, some conferred a high risk of cancer[31–33] whereas others conferred a moderate risk[34–38] (Table 4). The specific variants in *APC* and *ATM* are only associated with an adult-onset CPS, had the specific variants been associated with childhood-onset CPS, these genes would have been listed above.

These 29 patients had 31 PVs in total, including seven frameshift, five nonsense, eight missense, and three splice-site variants. Three variants were larger deletions of at least one exon, one patient had UPD 11p, and four patients had trisomy 21.

Some CPSs occurred in several patients (Table 4). Two patients had PVs in two CPS genes: a patient with LCH had PVs in *BRCA2* and *AXIN2*; a patient with a malignant peripheral nerve sheath tumor had PVs in *NF1* and *PALB2*. All CPS PVs were monoallelic, except for one patient with biallelic PVs in *DDX41*.[39]

Of the 21 childhood-associated CPSs 18 (85.7%) had previously established links between genotype (e.g., trisomy 21 and leukemia) and cancers (Table 4). This study identified a PV in eight of the 21 (38.1%) pediatric CPS patients, whereas 13/21 (61.9%) patients had a previously established genetic predisposition syndrome (e.g., NF1 or trisomy 21). Such a connection was established through clinical diagnosis and/or genetic testing. Among these 21 patients, only one had a family member diagnosed with cancer before 18 years of age (Table 5).

**Table 3. Patients distributed according to sex, age at diagnosis, diagnosis, level of feedback and fulfillment of the Jongmans' and MIPOGGs criteria.**

| | N (%) |
|---|---|
| **Sex** | |
| Male | 121 (61.1%) |
| Female | 77 (38.9%) |
| **Age at diagnosis** | |
| 0–5 years | 104 (52.5%) |
| 6–10 years | 41 (20.7%) |
| 11–15 years | 39 (19.7%) |
| 16–18 years | 14 (7.0%) |
| **Diagnosis** | |
| Hematologic cancer | 105 (53.0%) |
| Precursor B-ALL | 45 |
| Lymphoma | 22 |
| AML/CML/other myeloid leukemia | 17 |
| Precursor T-ALL | 10 |
| Langerhans cell histiocytosis | 6 |
| Myelodysplastic syndrome | 3 |
| Mixed lineage ALL | 2 |
| Tumors of the central nervous system | 44 (22.2%) |
| Low-grade glioma, WHO grade I–II | 17 |
| High-grade glioma, WHO grade III–IV | 6 |
| Ependymoma | 4 |
| AT/RT | 3 |
| Medulloblastoma | 2 |
| Schwannoma | 2 |
| Other | 10 |
| Solid tumors | 49 (25.0%) |
| Wilms tumor | 8 |
| Neuroblastoma | 8 |
| Rhabdomyosarcoma | 6 |
| Osteosarcoma | 5 |
| Retinoblastoma | 5 |
| Ewing's sarcoma | 4 |
| Malignant peripheral nerve sheath tumor | 1 |
| Other | 12 |
| **Level of feedback** | |
| Full feedback | 155 (78.2%) |
| Limited feedback | 24 (12.1%) |
| No feedback | 19 (9.6%) |
| **Fulfillment of Jongmans' criteria** | |
| Fulfilled one or more criteria | 56 (28.3%) |
| Did not fulfill any criteria | 142 (71.7%) |
| **Referral for genetic evaluation recommended by MIPOGG** | |
| Referral recommended | 64 (32.3%) |
| Referral not recommended | 134 (67.7%) |

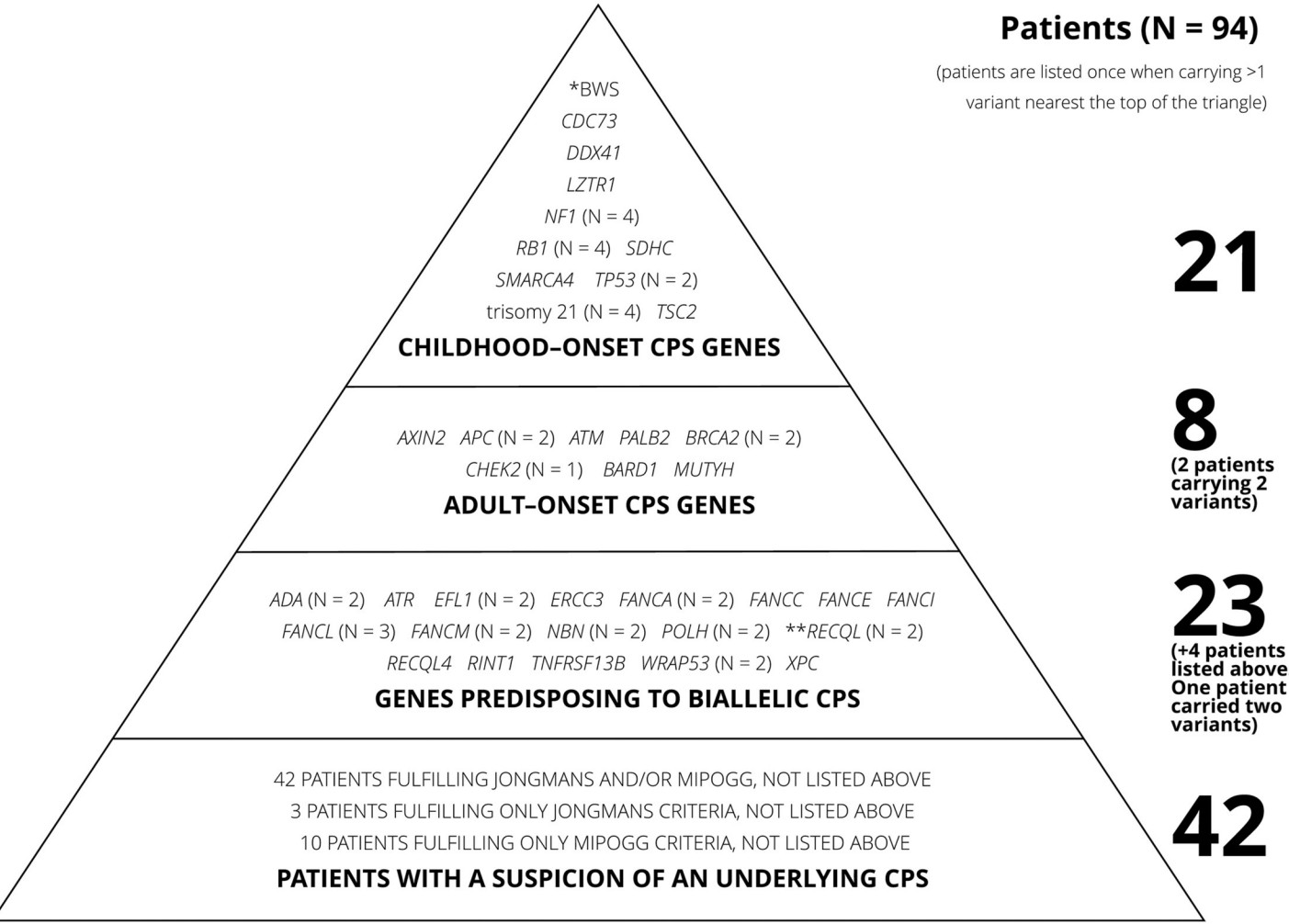

**Fig 2. Triangle of patients with confirmed or suspected underlying cancer predisposition syndrome.** Patients fulfilling criteria for more than one level of the triangle were only counted once, closest to the top of the triangle. The column on the left shows the number of pathogenic variants on each level. The column on the right shows the number of patients on each level.

### PVs in biallelic CPS

Twenty-seven of 198 (13.6%) patients carried one ($n = 22$) or two ($n = 1$, *FANCM* and *ADA*) PVs predisposing toward CPS through biallelic inheritance, of whom four also had a PV in a monoallelic CPS gene (Table 6). Variants were found in *ADA*, *ATR*, *EFL1*, *ERCC3*, *FANCA/C/ E/I/L/M*, *NBN*, *POLH*, *RECQL*, *RECQL4*, *RINT1*, *WRAP53*, and *XPC* (Fig 2).

Seven of the 198 (3.5%) patients carried two or more PVs/trisomy 21 (Table 7). Of these seven patients, one carried variants bioinformatically predicted to be oligogenically pathogenic. This patient carried biallelic variants in *DDX41* and a monoallelic variant in *NBN*. The digenic combination of a single *DDX41* variant and the *NBN* variant were also predicted to be pathogenic. No variant combinations in the six remaining patients were predicted to be oligogenically pathogenic.

### Variants of unknown significance

All 198 patients carried VUS in one or more CPS genes. VUS with a frequency <1% and a CADD/PHRED score >20 are listed in S1 Data. Thirty-nine of 198 (19.7%) patients had a

**Table 4. Pathogenic and likely pathogenic variants in cancer predisposition genes.**

| Diagnosis | Gene | Variant | Protein | Corresponding to phenotype of the patient | Inherited (M/P)[1] / de novo | CADD-PHRED[2] score |
|---|---|---|---|---|---|---|
| **Pathogenic/likely pathogenic variants (childhood onset)** | | | | | | |
| Acute myeloid leukemia | CDC73 | c.358C>T | p.(Arg120*) | – | – | 37.0 |
| Plasmacytoid dendritic cell leukemia | DDX41[3] DDX41 | c.962C>T c.937G>C | p.(Pro321Leu) p.(Gly313Arg) | + + | M P | 31.0 32.0 |
| Osteosarcoma | LZTR1 | c.955C>T | p.(Gln319*) | – | – | 38.0 |
| Optic nerve glioma | NF1 | c.5242C>T | p.(Arg1748*) | + | M | 38.0 |
| Acute myeloid leukemia | NF1 | c.288+1delG | p.(?) | + | – | 34.0 |
| Malignant peripheral nerve sheath tumor[4] | NF1 | c.61-22171_4110+2458del (deletion of exon 2–30) | p.(Leu21_Gln1370del) | + | M | – |
| Pilocytic astrocytoma | NF1 | c.2033dupC | p.(Ile679Aspfs*21) | + | – | 34.0 |
| Retinoblastoma | RB1 | c.1735C>T | p.(Arg579*) | + | – | 36.0 |
| Retinoblastoma | RB1 | c.409_412delGAAA | p.(Glu137Leufs*15) | + | P | 36.0 |
| Retinoblastoma | RB1 | c.219_220delAG | p.(Arg73Serfs*36) | + | De novo | 31.0 |
| Retinoblastoma | RB1 | c.2663+2T>C | p.(?) | + | – | 25.2 |
| Acute promyelocytic leukemia | SDHC | c.148C>T | p.(Arg50Cys) | – | – | 34.0 |
| Small cell carcinoma[5] | SMARCA4[6] | c.2002-625_2124-1045del (deletion of exon 14) | p.(Glu669Cysfs*7) | + | – | |
| Precursor B-cell acute lymphoblastic leukemia | TP53 | c.637C>G | p.(Arg213Gly) | + | P | 28.4 |
| Osteosarcoma | TP53 | c.818G>A | p.(Arg273His) | + | De novo | 27.3 |
| Subependymal giant cell astrocytoma | TSC2 | c.4141dupC | p.(Leu1382Profs*32) | + | De novo | 23.5 |
| Wilms tumor[7] | Paternal uniparental disomy of chromosome 11, corresponding to Beckwith–Wiedemann syndrome | | | + | P | – |
| Acute megakaryoblastic leukemia[8] | 47,XY,+21 | | | + | – | – |
| Acute megakaryoblastic leukemia[8] | 47,XY,+21 | | | + | – | – |
| Hodgkins lymphoma | 47,XX,+21 | | | + | – | – |
| Acute lymphoblastic leukemia | 47,XX,+21 | | | + | – | – |
| **Pathogenic/likely pathogenic variants (adult onset)** | | | | | | |
| Wilms tumor | APC[9, 10] | c.3920T>A | p.(Ile1307Lys) | – | – | 5.3 |
| Neuroblastoma | APC[9, 10] | c.3920T>A | p.(Ile1307Lys) | – | – | 5.3 |
| Rhabdomyosarcoma | ATM[9, 10] | c.9023G>C | p.(Arg3008Pro) | – | – | 32.0 |
| Wilms tumor | BARD1[10] | g.215591264-215774591 (Deletion of exon 1–11) | p.(?) | – | – | – |
| Langerhans cell histiocytosis | BRCA2 AXIN2 | c.5722_5723delCT c.815+1G>A | p.(Leu1908Argfs*2) p.(?) | – – | M M | 21.2 24.5 |
| T-cell acute lymphoblastic leukemia | BRCA2 | c.6486_6489delACAA | p.(Lys2162Asnfs*5) | – | – | 29.1 |
| T-cell acute lymphoblastic leukemia | CHEK2[10] | c.1100del | p.(Thr367Metfs*15) | – | – | 35.0 |
| Precursor B-cell acute lymphoblastic leukemia | MUTYH[10] | c.536A>G | p.(Tyr179Cys) | – | – | 24.7 |

(Continued)

**Table 4.** (Continued)

| Diagnosis | Gene | Variant | Protein | Corresponding to phenotype of the patient | Inherited (M/P)[1] / de novo | CADD-PHRED[2] score |
|---|---|---|---|---|---|---|
| Pathogenic/likely pathogenic variants (childhood onset) | | | | | | |
| Malignant peripheral nerve sheath tumor[4] | PALB2 | c.2736G>A | p.(Trp912*) | – | P | 41.0 |

– Value not known/applicable (e.g., parents not tested)

[1] M: maternally inherited, P: paternally inherited

[2] CADD: combined annotation dependent depletion, PHRED: quality score

[3] Putative childhood CPS gene

[4] Same patient carrying the NF1 deletion and PALB2 mutation

[5] The initial diagnosis (synovial sarcoma) was revised later revised

[6] Validation in process

[7] Detected by clinical analysis

[8] These patients are twin brothers

[9] The specific variants in APC and ATM are only associated with an adult-onset cancer predisposition syndrome; had the variant been associated with childhood cancer predisposition syndrome, these genes would have been listed above.

[10] These variants confer a moderate risk of cancer in adulthood

VUS in DNA repair pathway genes (ATM, BLM, NBN, MLH1, MSH2, MSH6, ERCC4, FANCA/E/F/G/I/L, BRCA1, BRCA2, RAD51C, RFWD3, SLX4), 16/198 (8.1%) patients had a VUS in genes associated with bone marrow failure (RPL5, RPS10, RPS19, CTC1, DKC1, NHP2, PARN, RTEL1, TERT, SBDS), 12/198 (6.1%) patients had a VUS in the RAS pathway (CBL, NF1, A2ML1, MAP2K2, PTPN11, RAF1, RRAS, SHOC2), and nine of 198 (4.5%) patients had a VUS in genes associated with familial leukemia (ETV6, GATA2, IKZF1, RUNX1, PAX5). Thirty-eight of the 198 (19.2%) patients had a VUS in a gene previously associated with the cancer diagnosed in the child. When subgroups were separated by diagnoses, 30 of 105 (28.6%) patients with hematologic malignancies had a VUS in a relevant gene associated with childhood onset (A2ML1, ADA, ATM, BLM, BRCA2, CREBBP, DDX41, DNAJC21, EFL1, EP300, ERCC6L2, FANCF, MAP2K2, PARN, PAX5, PTPN11, RPL5, RRAS, SH2D2A, SHOC2, SOS2, TERT, TNFRSF13B). One of the 44 (2.3%) patients with central nervous system (CNS) tumors had a VUS in a gene predisposing to Fanconi Anemia in which childhood-onset brain tumors have been described (FANCG), and seven of the 49 (14.3%) patients with solid tumors had a VUS in a relevant gene associated with childhood onset (BRCA2, CDH23, EP300, ERCC6L2, FANCI, RB1, RFWD3, XRCC2). Most variants occurred in DNA repair pathway genes (S1 Data). Four patients had a VUS, but none had a PV, in a mismatch repair pathway gene.

## Fulfillment of clinical criteria indicating an underlying CPS

All patients were evaluated using a phenotype checklist developed for this study (S1 Text), and 116/198 (58.6%) patients had one or more CPS-associated findings.

Overall, 70/198 (35.4%) patients fulfilled Jongmans' ($n = 56$, 28.3%) and/or MIPOGG criteria ($n = 64$, 32.3%) including 17 (81.0%) of the 21 patients with a childhood-onset CPS (Tables 8 and 9). Of the four patients with a PV in a childhood-onset CPS gene who did not fulfill Jongmans' criteria, none had excessive chemotherapy-induced toxicity. Patients not identified by either tool included two with PVs in TP53, and one with a pathogenic SMARCA4 variant. The patient with a PV in CDC73 was identifies by MIPOGG and not by Jongmans. The SMARCA4-deletion patient was diagnosed with synovial sarcoma of the ovary, revised to

**Table 5. Family history of patients with pathogenic and likely pathogenic variants in childhood-onset and adult-onset cancer genes.**

| Patient diagnosis | Gene | Variant | Protein | Family history |
|---|---|---|---|---|
| | | | **Patients with childhood-onset CPS** | |
| Acute myeloid leukemia | CDC73 | c.358C>T | p.(Arg120*) | 2nd-degree relative (maternal side): grandfather: chronic lymphoblastic leukemia, 62 years |
| Plasmacytoid dendritic cell leukemia | DDX41 DDX41 | c.962C>T c.937G>C | p.(Pro321Leu) p.(Gly313Arg) | No cancer cases in 1st–2nd-degree relatives Sister: intellectual disability corresponding to the phenotype of the syndrome identified in this patient |
| Osteosarcoma | LZTR1 | c.955C>T | p.(Gln319*) | 2nd-degree relative (maternal side): grandfather: lymphoma, 68 years |
| Optic nerve glioma | NF1 | c.5242C>T | p.(Arg1748*) | No cancer cases in 1st–2nd-degree relatives |
| Acute myeloid leukemia | NF1 | c.288+1delG | p.(?) | No cancer cases in 1st–2nd-degree relatives |
| Malignant peripheral nerve sheath tumor[1] | NF1 | c.61-22171_4110+2458del (deletion of exon 2–30) | p. (Leu21_Gln1370del) | No cancer cases in 1st–2nd-degree relatives |
| Pilocytic astrocytoma | NF1 | c.2033dupC | p.(Ile679Aspfs*21) | 2nd-degree relative (maternal side): grandfather: colon cancer, 84 years |
| Retinoblastoma | RB1 | c.1735C>T | p.(Arg579*) | No cancer cases in 1st–2nd-degree relatives |
| Retinoblastoma | RB1 | c.409_412delGAAA | p.(Glu137Leufs*15) | 1st-degree relative: father, retinoblastoma, 0 years 2nd-degree relatives (paternal side): father's sister: retinoblastoma (0 years), rhabdomyosarcoma (14 years), melanoma (20 years). Grandfather: melanoma (39 years), myelofibrosis (48 years) |
| Retinoblastoma | RB1 | c.219_220delAG | p.(Arg73Serfs*36) | No cancer cases in 1st–2nd-degree relatives |
| Retinoblastoma | RB1 | c.2663+2T>C | p.(?) | No cancer cases in 1st–2nd-degree relatives |
| Acute promyelocytic leukemia | SDHC | c.148C>T | p.(Arg50Cys) | No cancer cases in 1st–2nd-degree relatives |
| Small cell carcinoma | SMARCA4 | c.2002-625_2124-1045del (Deletion of exon 14) | p.(Glu669Cysfs*7) | 2nd-degree relative (paternal side): grandfather: lung cancer, 71 years, (maternal side): grandfather: lung cancer, 79 years |
| Precursor B-cell acute lymphoblastic leukemia | TP53 | c.637C>G | p.(Arg213Gly) | 1st-degree relative: father: pheochromocytoma, 34 years (not diagnosed at the time of the child's diagnosis) 2nd-degree relative (paternal side): grandmother: hepatic cholangiocarcinoma, 36 years |
| Osteosarcoma | TP53 | c.818G>A | p.(Arg273His) | 2nd-degree relative (paternal side): grandfather: colon cancer, 84 years |
| Subependymal giant cell astrocytoma | TSC2 | c.4141dupC | p.(Leu1382Profs*32) | 2nd-degree relative (maternal side): grandfather: urothelial carcinoma, 71 years |
| Wilms tumor[2] | Paternal uniparental disomy of chromosome 11, corresponding to Beckwith–Wiedemann syndrome | | | 2nd-degree relatives (maternal side): grandmother: lung cancer, 64 years, mother's brother: urothelial carcinoma, 41 years |
| Acute megakaryoblastic leukemia (two patients, twins) | 47,XY,+21 | | | 2nd-degree relative (paternal side): grandmother: lung cancer, 50 years |
| Hodgkin lymphoma | 47,XX,+21 | | | 2nd-degree relatives (maternal side): grandmother: ovarian cancer, 68 years, (paternal side): grandmother: gastrointestinal stromal tumor, 82 years |
| Precursor B-cell acute lymphoblastic leukemia | 47,XX,+21 | | | 2nd-degree relative: mother's sister: melanoma, 37 years |
| | | | **Patients with adult-onset CPS** | |
| Wilms tumor | APC | c.3920T>A | p.(Ile1307Lys) | 2nd-degree relative (paternal side): grandmother: urothelial carcinoma, 74 years |
| Neuroblastoma | APC | c.3920T>A | p.(Ile1307Lys) | No cancer cases in 1st–2nd-degree relatives |
| Rhabdomyosarcoma | ATM | c.9023G>C | p.(Arg3008Pro) | 2nd-degree relative (maternal side): mother's brother: tumor on heart valve, 0 years |
| Wilms tumor | BARD1 | g.215591264-215774591 (Deletion of exon 1–11) | p.(?) | No cancer cases in 1st–2nd-degree relatives |
| Langerhans cell histiocytosis | BRCA2 AXIN2 | c.5722_5723delCT c.815+1G>A | p.(Leu1908Argfs*2) p.(?) | 2nd-degree relative (paternal side): grandfather: esophageal cancer, 59 years |

*(Continued)*

**Table 5.** (Continued)

| Patient diagnosis | Gene | Variant | Protein | Family history |
|---|---|---|---|---|
| **Patients with childhood-onset CPS** | | | | |
| T-cell acute lymphoblastic leukemia | BRCA2 | c.6486_6489delACAA | p.(Lys2162Asnfs*5) | 2nd-degree relative (maternal side): grandmother: breast cancer, 63 years |
| T-cell acute lymphoblastic leukemia | CHEK2 | c.1100del | p.(Thr367Metfs*15) | 2nd-degree relatives (paternal side): grandfather: prostate cancer, 65 years, grandmother: cervical cancer, 54 years |
| Precursor B-cell acute lymphoblastic leukemia | MUTYH | c.536A>G | p.(Tyr179Cys) | No cancer cases in 1st–2nd-degree relatives |
| Malignant peripheral nerve sheath tumor[1] | PALB2 | c.2736G>A | p.(Trp912*) | No cancer cases in 1st–2nd-degree relatives |

[1]Same patient carrying these two variants.

[2]Not identified by whole-genome sequencing.

small-cell carcinoma based on this study, and would have fulfilled both criteria if the initial diagnosis had been correct. Of the patients with adult-onset CPS, 2/9 (22.2%) fulfilled Jongmans' ($n = 2$) and MIPOGG ($n = 1$) criteria, which is significantly fewer than for childhood-onset CPS ($p = 0.0022$). Of the additional 23 patients with a heterozygous PV predisposing to biallelic CPS, eight (34.8%) fulfilled Jongmans' ($n = 5$) and MIPOGG ($n = 7$) criteria.

The number of VUS identified were higher among patients without a CPS and with adult-onset CPS compared to patients with a childhood-onset CPS, in the first group patients on average carried 2.5 VUS compared to 1.6 VUS in the latter group. The same was the case when comparing patients with a childhood-onset CPS to patients who solely fulfilled Jongmans/MIPOGGs criteria, patients carrying a childhood-onset CPS carried an average of 1.6 VUS compared to 2.5 VUS among patients fulfilling Jongmans/MIPOGGs criteria alone.

## Family histories of cancer

Parents reported cancer diagnoses for 704 family members, 106 of whom resided outside Denmark, precluding further verification. Cancer diagnoses were verified for 328 (54.8%) of the remaining 598 family members, whereas the others did not consent to retrieval of medical records ($n = 45$) or their diagnoses could not be verified ($n = 225$) due to difficulties identifying distant/deceased family members or cancer occurrence prior to registration in Danish registries (before 1943). For 1st-, 2nd-, and 3rd-degree relatives 16 (100.0%), 133 (84.1%), and 179 (42.2%) cases were verified, respectively. The following is based on verified diagnoses and family recollection (for 1st to 3rd generation family members) when verification was impossible.

In total, 191/198 (96.5%) participants had a family history of cancer. Seven of 198 (3.5%) participants had a family member diagnosed with cancer before 18 years of age (two had a CPS). Fifty-six of 198 (28.3%) participants had at least one relative diagnosed with cancer between the ages of 18 and 45. Three of 198 (1.5%) participants had two or more relatives under the age of 45 diagnosed with cancer.

Forty-three of 198 (21.7%) participants had a relative with a cancer of the same organ system as the patient. Patients with hematologic malignancies and solid tumors did not have more family members with cancers of the same organ system than the other two patient groups. In contrast, patients with a CNS tumor had a family member with a malignancy in the CNS more frequently than patients with either solid tumors or hematologic malignancies ($p = 0.04$). This association also held ($p = 0.04$) when patients with a CPS were eliminated (Table 10). Family history for the patients with a confirmed CPS can be found in Table 5.

**Table 6. Patients with pathogenic or likely pathogenic variants in biallelic cancer predisposition genes.**

| Diagnosis | Gene | Variant | Protein | CADD-PHRED score |
|---|---|---|---|---|
| Acute myeloid leukemia[1] | ADA | c.646G>A | p.(Gly216Arg) | 28.0 |
| Precursor B-cell acute lymphoblastic leukemia | ADA | c.1078+2T>A | p.(?) | 22.9 |
| Chronic myeloid leukemia | ATR | c.2320dupA | p.(Ile774Asnfs*3) | 20.1 |
| Rhabdomyosarcoma | EFL1 | c.159+3A>G | p.(?) | 14.84 |
| Precursor T-cell acute lymphoblastic leukemia | EFL1 | c.2430_2431delCC | p.(Leu811Asnfs*10) | 15.9 |
| Wilms tumor | ERCC3 | c.1115_1120dupAGCAGT | p.(Trp374*) | 37.0 |
| Astrocytoma | FANCA | c.3482C>T | p.(Thr1161Met) | 17.3 |
| Precursor B-cell acute lymphoblastic leukemia | FANCA | c.3391A>G | p.(Thr1131Ala) | 23.5 |
| Acute myeloid leukemia | FANCC | c.535C>T | p.(Arg179*) | 35.0 |
| Yolk sac tumor | FANCE | c.108delG | p.(Pro37Leufs*47) | 16.7 |
| Precursor B-cell acute lymphoblastic leukemia | FANCI | c.158-5A>G | p.(?) | 12.7 |
| Precursor T-cell acute lymphoblastic leukemia | FANCL | c.540+1G>A | p.(?) | 23.3 |
| Precursor B-cell acute lymphoblastic leukemia | FANCL | c.1007_1009delTAT | p.(Ile336_Cys337delinsSer) | 23.5 |
| Lymphoma | FANCL | c.1096_1099dupATTA | p.(Thr367Asnfs*13) | 35.0 |
| Acute myeloid leukemia[1] | FANCM | c.681+1G>C | p.(?) | 25.9 |
| Rhabdomyosarcoma | FANCM | c.2156_2160delAACCA | p.(Lys719Serfs*15) | 36.0 |
| Plasmacytoid dendritic cell leukemia | NBN | c.156_157delTT | p.(Ser53Cysfs*8) | 25.6 |
| Wilms tumor | NBN | c.834dupA | p.(Gln279Serfs*6) | 37.0 |
| Precursor B-cell acute lymphoblastic leukemia | POLH | c.1600_1610delCA | p.(Gln534Glufs*11) | 40.0 |
| Precursor B-cell acute lymphoblastic leukemia | POLH | c.491-69_660+30del (deletion of exon 5) | p.(Glu164Glyfs*37) | – |
| Ganglioglioma | RECQL | c.1859C>G | p.(Ser620*) | 38.0 |
| Craniopharyngioma | RECQL | c.1859C>G | p.(Ser620*) | 38.0 |
| Precursor B-cell acute lymphoblastic leukemia | RECQL4 | c.3072delA | p.(Val1026Cysfs*18) | 21.7 |
| Lymphoma | RINT1 | c.88+3A>G | p.(?) | 15.4 |
| Glioma[2] | TNFRSF13B | c.431C>G | p.(Ser144*) | 35.0 |
| Craniopharyngioma | WRAP53 | c.1192C>T | p.(Arg398Trp) | 34.0 |
| Precursor B-cell acute lymphoblastic leukemia | WRAP53 | c.1192C>T | p.(Arg398Trp) | 34.0 |
| Neurofibroma | XPC | c.1934delC | p.(Pro645Leufs*5) | 26.6 |

Three of the patients listed above also had a monoallelic pathogenic germline mutation (fam no. 13, 25 and 51).

Monoallelic variants in RECQL are pathogenic; however, their relationship to cancer is uncertain. Therefore, they are listed here.

[1]Same patient carrying the ADA and FANCM variants.

[2]Pathogenic variants (Romberg et al., 2013) may be inherited via an autosomal dominant or autosomal recessive. pattern. Based on the patient's phenotype, this variant was considered inherited recessively.

## Secondary findings

Two patients had PVs in genes associated with familial hypercholesterolemia (*APOB* and *LDLR*), three had PVs in genes associated with arrhythmic right ventricular cardiomyopathy (*DSC2*, *DSG2*, and *PKP2*), and one patient had a PV in *KCNQ1*, which is associated with long QT syndrome. These six variants are associated with increased risk of disease and families were informed. Overall, 18 patients (9.1%) had an ACMG 'actionable' PV, including 12 PVs in CPS genes and six PVs in genes associated with other non-malignant diseases (Table 11).

## Discussion

In this first nationwide unselected cohort of consecutive pediatric cancer patients, half had a likely or validated underlying CPS, based on WGS, clinical examination, and pedigree

**Table 7. Patients carrying more than one pathogenic/likely pathogenic variant.**

| Diagnosis of the patient | Gene | Variant | Protein | Gene | Variant | Protein | Gene | Variant | Protein |
|---|---|---|---|---|---|---|---|---|---|
| Malignant peripheral nerve sheath tumor | NF1 | c.61-22171_4110+2458del (deletion of exon 2–30) | p.(Leu21_Gln1370del) | PALB2 | c.2736G>A | p.(Trp912*) | – | – | – |
| Optic nerve glioma | NF1 | c.5242C>T | p.(Arg1748*) | XPC | c.1934delC | p.(Pro645Leufs*5) | – | – | – |
| B-cell acute lymphoblastic leukemia | 47,XX,+21 | | | POLH | c.1600_1610delCA | p.(Gln534Glufs*11) | – | – | – |
| Plasmacytoid dendritic cell leukemia | DDX41 | c.962C>T | p.(Pro321Leu) | DDX41 | c.937G>C | p.(Gly313Arg) | NBN | c.834dupA | p.(Gln279Serfs*6) |
| Wilms tumor | BARD1 | g.215591264-215774591 (deletion of exon 1–11) | p.(?) | ERCC3 | c.1115_1120dupAGCAGT | p.(Trp374*) | – | – | – |
| Langerhans cell histiocytosis | BRCA2 | c.5722_5723delCT | p.(Leu1908Argfs*2) | AXIN2 | c.815+1G>A | p.(?) | – | – | – |
| Acute myeloid leukemia | ADA | c.646G>A | p.(Gly216Arg) | FANCM | c.681+1G>C | p.(?) | – | – | – |

mapping, and 14.6% had a genetically verified CPS. These findings strongly indicate that genetic predisposition to childhood cancer may be far more common than previously supposed. Furthermore, other modes of inheritance (di-, oligo- and polygenic risk) may play significant roles in pathogenesis.

Our childhood-onset CPS results are consistent with previous studies that found a CPS in 7–10% of pediatric cancer patients.[8–11]

The frequency of PVs in our study was significantly higher than that observed in control cohorts wherein 0.6–1.1% of adult patients in a Genomics England cohort and a cohort of pediatric and adult patients with autism had PVs in CPS genes.[7] This study, however, excluded patients with known CPS and included different genes (including genes with frequent somatic variants) and is thus not directly comparable. A study of osteosarcoma patients found a remarkably high frequency of CPSs in control cohorts (12.1% and 9.3%), probably because many of the genes included (e.g., PMS1, and COL7A1) were not definitely linked to cancer.[6] Other studies have included adult-onset CPS genes. For example, Zhang et al. found only 0.7% of their patients carried an adult-onset CPS variant (BRCA1/2 and PALB2).[7] We found that 1.5% of our patients carried PVs in BRCA2 and PALB2. Interestingly, another Scandinavian study reported a significantly higher prevalence of childhood cancer in families with PVs in BRCA2.[40] Similarly, Wilson et al. showed that BRCA2 was one of the most frequently mutated genes among childhood cancer survivors.[11] Therefore, BRCA2 variants may be important in childhood cancer etiology,[40] potentially influencing treatment options if deficiencies in homologous repair promote some tumors. Even though a higher frequency of PVs in BRCA2 than BRCA1 have been found in the general population[41], this does not explain why multiple studies have identified so few PVs in BRCA1, which, in the Danish population, are more frequently associated with breast cancer than PVs in BRCA2.[42,43] Overall, WGS data from ethnically comparable children are lacking making comparisons of genetic findings in patients difficult. As most children survive cancer; therefore, identifying adult-onset CPSs is important for future surveillance and counselling.

**Table 8. Patients with an underlying cancer predisposition syndrome according to Jongmans' and MIPOGG criteria.**

| Diagnosis (patient ID) | CPS (gene) | Jongmans | MIPOGG |
|---|---|---|---|
| Precursor B-cell acute lymphoblastic leukemia (A1) | Li–Fraumeni syndrome (TP53) | – | – |
| Osteosarcoma (A2) | Li–Fraumeni syndrome (TP53) | – | – |
| Acute promyelocyte leukemia (B1) | Familial paraganglioma and pheochromocytoma syndrome (SDHC) | + | + |
| Acute myeloid leukemia (C1) | Hyperparathyroidism-Jaw tumor syndrome (CDC73) | – | + |
| Acute myeloid leukemia (E1) | Neurofibromatosis type 1 (NF1) | + | + |
| Malignant peripheral nerve sheath tumor[1] (E2) | Neurofibromatosis type 1 (NF1) | + | + |
| Optic nerve glioma (E3) | Neurofibromatosis type 1 (NF1) | + | + |
| Pilocytic astrocytoma (E4) | Neurofibromatosis type 1 (NF1) | + | + |
| Osteosarcoma (F1) | Schwannomatosis/Noonan syndrome (LZTR1) | + | + |
| Plasmacytoid dendritic cell leukemia (G1) | Novel putative childhood leukemia cancer predisposition syndrome (biallelic DDX41) | + | + |
| Retinoblastoma (H1) | Familial retinoblastoma syndrome (RB1) | + | + |
| Retinoblastoma (H2) | Familial retinoblastoma syndrome (RB1) | + | + |
| Retinoblastoma (H3) | Familial retinoblastoma syndrome (RB1) | + | + |
| Retinoblastoma (H4) | Familial retinoblastoma syndrome (RB1) | + | + |
| Wilms tumor (I1) | Beckwith–Wiedemann syndrome (pUPD chr 11) | + | + |
| Subependymal giant cell astrocytoma (J1) | Tuberous sclerosis complex (TSC2) | + | + |
| Small cell carcinoma of the ovary (K1) | Rhabdoid tumor predisposition syndrome (SMARCA4) | – | – |
| Acute myeloid leukemia (L1) | Down syndrome (46,XY+21) | + | + |
| Acute myeloid leukemia (L2) | Down syndrome (46,XY+21) | + | + |
| Hodgkin lymphoma (L3) | Down syndrome (46,XX+21) | + | + |
| Precursor B-cell acute lymphoblastic leukemia (L4) | Down syndrome (46,XX+21) | + | + |
| Total number of patients with childhood cancer predisposition syndrome | | 17/21 = 80.9% | 18/21 = 85.7% |
| Precursor B-cell acute lymphoblastic leukemia (M1) | MUTYH–associated polyposis (MUTYH) | – | – |
| T-cell acute lymphoblastic leukemia (N1) | Familial breast and ovarian cancer (BRCA2) | – | – |
| T-cell acute lymphoblastic leukemia (O1) | Familial breast cancer (CHEK2) | – | – |
| Langerhans cell histiocytosis (N2) | Familial breast and ovarian cancer (BRCA2), oligodontia-colorectal cancer syndrome (AXIN2) | – | – |
| Malignant peripheral nerve sheath tumor [1] (P1) | Familial breast and ovarian cancer (PALB2) | + (fulfills due to NF1 variant, not counted below) | + (fulfills due to NF1 variant) |
| Neuroblastoma (Q1) | Familial adenomatous polyposis (APC) | + | – |
| Wilms tumor (Q2) | Familial adenomatous polyposis (APC) | + | + |
| Rhabdomyosarcoma (R1) | Ataxia telangiectasia (ATM) | – | – |
| Wilms tumor (S1) | Familial breast and ovarian cancer and familial neuroblastoma (BARD1) | – | – |
| Total number of patients with adult-onset cancer predisposition syndrome | | 2/9 = 22.2% | 1/9 = 11.1% |

+ fulfills the criteria,–does not fulfill the criteria.

[1]same patient carrying these variants, patient not counted in the adult-onset cancer predisposition syndrome.

Red: childhood cancer predispositions syndrome, Blue: adult-onset cancer predisposition syndrome.

**Table 9. Phenotypes identified using Jongmans'/MIPOGG criteria.**

| Cancer diagnosis | Non-cancer diagnosis | Phenotypic finding | Genetic findings |
|---|---|---|---|
| Diffuse intrinsic pontine glioma | Autism spectrum disorder | developmental delay, speech delay, learning difficulties | Klinefelter syndrome |
| Neuroblastoma | Autism spectrum disorder | developmental delay, learning difficulties | – |
| Acute myeloid leukemia | Autism spectrum disorder | developmental delay (does not speak until age four), learning difficulties, strabismus | – |
| Glioblastoma | Autism spectrum disorder | developmental delay (speech delay), learning difficulties | – |
| Acute myeloid leukemia: monozygotic twin of patient below | Down syndrome | intellectual disability, epicanthus, strabismus, developmental delay, single palmar crease | 47,XY,+21 |
| Acute myeloid leukemia: monozygotic twin of patient above | Down syndrome | intellectual disability, epicanthus, strabismus, developmental delay, single palmar crease | 47,XY,+21 |
| Hodgkin lymphoma | Down syndrome | intellectual disability, epicanthus, strabismus, developmental delay, single palmar crease | 47,XX,+21 |
| Precursor B-cell acute lymphoblastic leukemia | Down syndrome | intellectual disability, epicanthus, strabismus, flat-footed, hearing deficit | 47,XX,+21 *POLH*, c.1600_1610delCA, p.(Gln534Glufs*11) |
| Acute myeloid leukemia | Neurofibromatosis type 1 | multiple café-au-lait spots | *NF1*, c.288+1delG, p.(?) |
| Neurofibroma | Neurofibromatosis type 1 | multiple café-au-lait spots | *NF1*, c.5242C>T, p.(Arg1748*) *XPC*, c.1934delC, p.(Pro645Leufs*5) |
| Malignant peripheral nerve sheath tumor | Neurofibromatosis type 1 | multiple café-au-lait spots hearing deficits left ear | *NF1*, c.61-22171_4110+2458del (deletion of exon 2–30), p.(Leu21_Gln1370del) *PALB2*, c.2736G>A, p.(Trp912*) |
| Pilocytic astrocytoma | Neurofibromatosis type 1 | multiple café-au-lait spots near-sightedness | *NF1*, c.2033dupC, p.(Ile679Aspfs21) |
| Subependymal giant cell astrocytoma | Tuberous sclerosis | intellectual disability, hypomelanotic macules, seizures | *TSC2*, c.4141dupC, p.(Leu1382Profs*32) |
| T-cell acute lymphoblastic leukemia | Goldenhar syndrome, craniofacial microsomia | mild phenotype with skintags in front of both ears, lack of iris coloring in part of the right eye | *EFL1*, c.2430_2431delCC, p.(Leu811Asnfs*10) (no genes are known to cause Goldenhar syndrome) |
| Hodgkin lymphoma | Behcet's disease | frequent mucocutaneous ulcerations, hyperpigmentation of the lower back | – (tissue type: HLA-B7) |
| Precursor B-cell acute lymphoblastic leukemia | Suspicion of Charcot–Marie–Tooth, but no definite molecular genetic diagnosis | strabismus, delayed motor development, hypotonia, hyperpigmentation of the head and legs, severe vincristine toxicity. Mother has a form of skeletal dysplasia with short fingers and arms, extensions defect in the elbow joint, flat feet | *SH3TC2*, c.279G>A (AR) *KIF1B*, c.3401CT, p.(Pro1134Leu) (VUS, parental testing planned) |
| Wilms tumor | Beckwith–Wiedemann syndrome | epicanthus, down slanting palpebrae, hypertelorism, macroglossia, overgrowth | Paternal uniparental disomy of chromosome 11, corresponding to Beckwith–Wiedemann syndrome phenotype |
| Plasmacytoid dendritic cell leukemia | Novel childhood predisposition syndrome associated with leukemia and intellectual disability | macroglossia, poor mouth motor skills, small milk teeth, deformed fingers and toes, hypotonia | *DDX41*, c.962C>T, p.(Pro321Leu) *DDX41*, c.937G>C, p.(Gly313Arg) *NBN*, c.156_157delTT, p.(Ser53Cysfs*8) |
| T-cell acute lymphoblastic leukemia | No genetic diagnosis | deeply set eyes, hypertelorism, large café-au-lait spots, right leg | – |
| Hodgkin lymphoma | No genetic diagnosis | severe speech delay–speech not understandable age 5 years, macrocephaly | – |

– No relevant genetic variants.

Many of our patients carried PVs in genes involved in DNA repair. Children have high cell-division rates, and deficiencies in DNA repair may result in accumulation of DNA damage and ultimately cancer. Fanconi anemia (FA) is associated with many large genes, and the

**Table 10. Family pedigree findings for 1st–3rd degree relatives.**

| | All patients, $n$ (%) | Patients with pathogenic/likely pathogenic variant in cancer predisposition gene, $n$ (%)[1] |
|---|---|---|
| **Malignancies in family members <18 years** | **7 (3.5%)** | **2 (6.9%)** |
| Relatives with cancer aged 18–45 years | 55 (27.8%) | 9 (31.0%) |
| Two or more 1st- or 2nd-degree relatives in the same parental lineage with cancer <45 years | 4 (2.0%) | 1 (3.4%) |
| Any cancer history in the family | 189 (95.5%) | 29 (100.0%) |
| More than one family member with cancer | 174 (87.9%) | 28 (96.6%) |
| Any family member with cancer of the same organ as the patient | 33 (16.7%) | 7 (24.1%) |
| Any family member with a hematologic malignancy<br> - To a child with a hematologic tumor<br> - To a child with a CNS tumor<br> - To a child with a solid tumor | 43 (21.7%)<br>25 (23.8%)<br>8 (18.2%)<br>10 (20.4%) | 9 (31.0%) |
| Any family member with a CNS tumor<br> - a child with a hematologic tumor<br> - To a child with a CNS tumor<br> - To a child with a solid tumor | 25 (12.6%)<br>9 (8.6%)<br>11 (25.0%)<br>5 (10.2%) | 2 (6.9%) |
| Any family member with a solid tumor (defined as any kidney tumor, retinoblastoma, bone tumor, neuroendocrine tumors, gastrointestinal stromal tumor, or rhabdomyosarcoma)<br> - To a child with a hematologic tumor<br> - To a child with a CNS tumor<br> - To a child with a solid tumor | 27 (13.6%)<br>14 (13.3%)<br>6 (13.6%)<br>7 (14.3%) | 7 (24.1%) |
| Chi2 test for differences between the groups | $p = 0.04$ | |
| Any family member with breast cancer<br>Two or more family members with breast cancer | 76 (38.4%)<br>29 (14.6%) | 11 (37.9%)<br>2 (6.9%) |
| Any family member with any gastrointestinal cancer<br>Two or more family members with any gastrointestinal cancer | 74 (37.4%)<br>20 (10.1%) | 12 (41.4%)<br>4 (13.8%) |

[1]Percentages are fractions of the 29 patients with a cancer predisposition syndrome.

frequency of PVs in FA genes was 4.3% in an adult population of 7,578 patients from the Exome Sequencing Project and the 1000 Genomes Project.[44] This is consistent with our results, which showed that 13 (6.6%) patients carried a PV in a FA gene. Pathogenic FA variants are associated with a small increase in lifetime adult-onset cancer risk,[45–47] and this may also be true for childhood-onset CPS.

**Table 11. Pathogenic/likely pathogenic variants in genes deemed 'actionable' by the American College of Medical Genetics.**

| Phenotype | Diagnosis | Gene/chromosomal alteration | Variant | Protein | Events corresponding to carrier status |
|---|---|---|---|---|---|
| Familial hypercholesterolemia | Burkitt lymphoma | APOB | c.1013delC | p.(Gln3378Hisfs*4) | – |
| Familial hypercholesterolemia | Glioma | LDLR | c.409G>A | p.(Gly137Ser) | – |
| Arrhythmogenic right ventricular cardiomyopathy | Diffuse intrinsic pontine glioma | PKP2 | c.1643delG | p.(Gly548Valfs*15) | – |
| Arrhythmogenic right ventricular cardiomyopathy | Precursor B-cell acute lymphoblastic leukemia | DSC2 | c.2508+5G>A | p.(?) | – |
| Arrhythmogenic right ventricular cardiomyopathy | Glioma | DSG2 | c.918G>A | p.(Trp306*) | – |
| Romano–Ward long QT syndrome | Anaplastic large-cell lymphoma | KCNQ1 | c.905C>T | p.(Ala302Val) | Cardiac arrest during treatment, attributed to large tumor in thorax and subsequent mechanical obstruction |

– No known clinical events

Interestingly, we observed one *CDC73* VUS and one PV in two patients with hematologic malignancies. PVs in *CDC73* cause 'hyperparathyroidism-jaw tumor syndrome' and parathyroid carcinoma,[48] and have been linked with hematologic cancer in mouse models.[49] RNA sequencing of leukemic cells from these patients showed no LOH, making a causal association less likely but not impossible.[50] Furthermore, we identified two patients with heterozygous deleterious variants in *ERCC6L2*, a gene linked to a bone marrow failure syndrome. [51] These patients were diagnosed with T-lineage acute lymphoblastic leukemia (ALL) and rhabdomyosarcoma, respectively. Tumor tissue was not available for further investigation.

Seven patients had more than one PV in CPS genes, suggesting that di-, oligo-, and polygenic inheritance can cause predisposition to childhood cancer. Four of these patients exhibited the phenotype associated with the childhood-onset PV. Bioinformatic predictions suggested that pathogenicity was highly likely in one of the seven patients. Other studies have also found more than one PV in the same patient, suggesting that digenic/polygenic inheritance may play a role in childhood cancer etiology,[8,52,53] as *MUTYH* and *OGG1* do in colorectal cancer etiology.[54] Kuhlen *et al.*[55] proposed a model of concomitant digenic inheritance involving two PVs within the same pathway combining to increase the likelihood of disease development. Generally, the observations from this and other studies suggest that the risk of disease development may increase by having more than one PV, even if the corresponding genes function in different pathways.

PVs in genes not previously associated with cancer development were identified. Some of these genes were 'actionable' and their identities were disclosed to the children's families, in accordance with ACMG recommendations. It is important to identify genes associated with increased risk of cardiac disease in pediatric cancer patients due to the increased risk of both cardiomyopathy[56] and symptoms in patients with long QT syndrome[57] undergoing anticancer treatment. PVs in these genes may also have clinical implications for family members.

We applied Jongmans' and MIPOGG criteria to assess the risk of underlying childhood CPSs. The majority of patients (85.7%) with a childhood-onset CPS were identified using these criteria. However, among the three unidentified patients were the two with Li-Fraumeni syndrome (one with ALL and one with osteosarcoma). Li-Fraumeni syndrome is associated with a high lifetime-risk of cancer, and the risk of a secondary cancer is further increased when the first cancer occurs during childhood.[58,59] Data suggest that surveillance programs for Li-Fraumeni patients increase their survival rates.[60] However, in contrast to other studies, patients with Li-Fraumeni syndrome were not identified here.[8,16] A possible explanation is that one of our patients carried a *de novo TP53* variant that could not be identified from a family history of cancer. Additionally, CPSs that are not associated with syndromic features may not fulfill relevant criteria and will be difficult to identify if the cancer is not pathognomonic of the CPS. Family history was only rarely the cause of fulfillment of Jongmans'/MIPOGG criteria in both patients with childhood- and adult-onset CPS. The primary causes of fulfillment of Jongmans'/MIPOGG criteria were the patient's diagnosis and clinical characteristics. This is interesting as family history is believed to be highly indicative of adult-onset CPS like hereditary breast- and ovarian cancer and Lynch syndrome. A possible explanation for this could be that more variants are *de novo* in pediatric cancer patients and that the age of pediatric cancer patient's parents is lower than parents of adult cancer patients.

We found that a family history of CNS tumors was associated with the case of childhood CNS tumors. However, only two of 44 patients with a CNS tumor carried a germline variant (*TSC2*, *NF1*), and none of these two patients had a 1$^{st}$–3$^{rd}$-degree family member with a CNS tumor. Therefore, there may be unidentified predisposition genes among CNS tumor patients. However, recall bias cannot be excluded, because CNS tumors among family members might be more memorable, especially if a child is diagnosed with a CNS tumor.

One limitation of this study is the lack of a comparison cohort, because population-based WGS data from ethnically comparable children are not publicly available. Another problem is whether PVs in cancer genes of children with cancers that are unassociated with that particular gene have occurred randomly. Other studies have found similar cases, in which the genotype and phenotype were not previously reported,[4,8] and it remains uncertain whether PVs in adult CPSs are driver or passenger mutations.[5,61] Thus, a large international collection of cases should be investigated to describe the phenotypic spectrum associated with each CPS variant.

Strengths of our study include the national setup with a consecutive cohort of unselected pediatric cancer patients, in-depth clinical examinations of children with cancer, and use of national databases to verify cancer diagnoses in family members. Additionally, we performed WGS instead of WES or gene panel analyses so that large structural rearrangements and CNVs could be identified, if present. Moreover, WGS will facilitate future analyses of deep intronic variants that affect splicing, variants within putative regulatory areas, and novel CPS genes.

These results demonstrate the value of systematically screening pediatric cancer patients for CPSs and strongly indicate that a higher proportion of childhood cancers may be linked to predisposing germline variants than previously supposed.

## Supporting information

**S1 Data. Gene panel and List of Variants of Unknown Significance (VUS).** Gene panel outlining the 314 genes examined in each patient included in this study. List outlining all the VUS (CADD-PHRED score >20 and allele frequency <1%) found in the patients included in this study.
(XLSX)

**S1 Text. Clinical checklist used in the clinical examination of all patient.**
(DOCX)

## Acknowledgments

We should like to acknowledge the research nurses in all Pediatric Oncology Departments in Denmark, they were instrumental in making this study possible.

## Author Contributions

**Conceptualization:** Kjeld Schmiegelow, Karin Wadt.

**Data curation:** Anna Byrjalsen, Thomas V. O. Hansen, Anne–Marie Gerdes, Kjeld Schmiegelow, Karin Wadt.

**Formal analysis:** Anna Byrjalsen, Thomas V. O. Hansen, Mana M. Mehrjouy, Karin Wadt.

**Funding acquisition:** Anna Byrjalsen, Ulrik K. Stoltze, Rasmus L. Marvig, Anne–Marie Gerdes, Kjeld Schmiegelow, Karin Wadt.

**Investigation:** Anna Byrjalsen, Lisa L. Hjalgrim, René Mathiasen, Charlotte K. Lautrup, Pernille A. Gregersen, Henrik Hasle, Peder S. Wehner, Ruta Tuckuviene, Maria Rossing, Rasmus L. Marvig, Niels Tommerup, Tina Elisabeth Olsen, David Scheie, Kjeld Schmiegelow, Karin Wadt.

**Project administration:** Kjeld Schmiegelow, Karin Wadt.

**Software:** Ulrik K. Stoltze, Nanna Moeller Barnkob, Peter Wad Sackett, Adrian O. Laspiur, Ramneek Gupta, Karin Wadt.

**Supervision:** Anne–Marie Gerdes, Kjeld Schmiegelow, Karin Wadt.

**Writing – original draft:** Anna Byrjalsen.

**Writing – review & editing:** Anna Byrjalsen.

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
