## [Decision Letter · Decision Letter 0]

28 Sep 2020

Dear Dr Byrjalsen,

Thank you very much for submitting your Research Article entitled 'Nationwide Germline Whole Genome Sequencing of 198 Consecutive Pediatric Cancer Patients Reveals a High Incidence of Cancer Prone Syndromes' to PLOS Genetics. Your manuscript was fully evaluated at the editorial level and by independent peer reviewers. The reviewers appreciated the attention to an important topic but identified some aspects of the manuscript that should be improved.

We therefore ask you to modify the manuscript according to the review recommendations before we can consider your manuscript for acceptance. Your revisions should address the specific points made by each reviewer.

[LINK]

Yours sincerely,

Charis Eng

Associate Editor

PLOS Genetics

Peter McKinnon

Section Editor: Cancer Genetics

PLOS Genetics

Please address the comments of both expert reviewers.

Reviewer's Responses to Questions

**Comments to the Authors:**

Reviewer #1: The review note is uploaded

Reviewer #2: Byrjalsen et al submitted a manuscript summarizing the results of nationwide germline whole genome sequencing in 198 consecutive pediatric cancer patients over two years (July 2016 to June 2018) from 4 pediatric cancer units in Denmark. Main results of this study include the discovery of molecularly confirmed cancer predisposition syndrome (CPS) or suspected CPS based on Jongmans’/MIPOGG criteria in close to half of the study cohort. This study replicated results of prior study (Zang et al) that childhood-onset CPS can be seen in up to 10% of newly diagnosed pediatric cancers. The authors claim the strength of their study methods to include broader range pediatric cancer diagnosis and the inclusion of structural variants detection in their genomic analysis pipeline. This study also validated the sensitivity of MIPOGG and Jongmans’ clinical criteria as screening tools for childhood-onset CPS.

Major comments and questions:

1. The authors identified 29/94 (31%) had molecularly confirmed CPS leaving the other 65 patients met the only clinical criteria for CPS. It will be interesting to do subset analysis on this group of patients as far as the number of VUS and enrichment of pathways compared to those who screened negative.

2. DDX41 was listed as gene predisposing to childhood cancers. To this reviewer knowledge, there has no reported cases of cancers in pediatric group (0-17 years) linked to pathogenic germline DDX41 variants outside of this study. Please provide reference(s) to support your argument.

3. Germline mutations in APC has been linked to pediatric medulloblastoma so this gene should be listed as gene associated with childhood and adult onset CPS.

4. The authors reported that the study identified a pathogenic variant in 8 out of 21 pediatric CPS patients while the other 13 were previously known to have CPS. Please clarify how these 13 patients were ascertained to have CPS prior to this study?

5. Under the family history section, I would advise describing the status of family history of cancers in 29 cases with confirmed pathogenic variants in CPS genes.

6. Only 22% of the patients with adult-onset CPS screened positive on Jongmans’/MIPOGG criteria compared to 86 % for childhood-onset CPS. It will be a great point for discussion to see if this discrepancy is due to family history status or unique type of cancers between the two groups.

Minor comments:

1. It’s easier for the readers to follow the manuscript when a proportion is written in a uniform way (listing nominator and denominator), for instance : “Overall 70/198 (35.4%) patients fulfilled….”

2. Page 11 last paragraph : “(85.7%)” should be placed after “18”

**Have all data underlying the figures and results presented in the manuscript been provided?**

Reviewer #1: Yes

Reviewer #2: Yes

PLOS authors have the option to publish the peer review history of their article (what does this mean?). If published, this will include your full peer review and any attached files.

Reviewer #1: No

Reviewer #2: No

---

## [Editor Report · Decision Letter 1]

28 Oct 2020

Dear Dr Byrjalsen,

We are pleased to inform you that your manuscript entitled "Nationwide Germline Whole Genome Sequencing of 198 Consecutive Pediatric Cancer Patients Reveals a High Incidence of Cancer Prone Syndromes" has been editorially accepted for publication in PLOS Genetics. Congratulations!

Yours sincerely,

Charis Eng

Associate Editor

PLOS Genetics

Peter McKinnon

Section Editor: Cancer Genetics

PLOS Genetics

Comments from the reviewers (if applicable):

Thank you for your responsiveness to original review.

**Data Deposition**

http://datadryad.org/submit?journalID=pgenetics&manu=PGENETICS-D-20-01316R1

**Press Queries**

---

## [Editor Report · Acceptance letter]

7 Dec 2020

PGENETICS-D-20-01316R1 

Nationwide Germline Whole Genome Sequencing of 198 Consecutive Pediatric Cancer Patients Reveals a High Incidence of Cancer Prone Syndromes 

Dear Dr Wadt, 

We are pleased to inform you that your manuscript entitled "Nationwide Germline Whole Genome Sequencing of 198 Consecutive Pediatric Cancer Patients Reveals a High Incidence of Cancer Prone Syndromes" has been formally accepted for publication in PLOS Genetics! Your manuscript is now with our production department and you will be notified of the publication date in due course.

With kind regards,

Nicola Davies

PLOS Genetics

On behalf of:
